# Spermidine/Spermine N1-Acetyltransferase 1 (*SAT1*)—A Potential Gene Target for Selective Sensitization of Glioblastoma Cells Using an Ionizable Lipid Nanoparticle to Deliver siRNA

**DOI:** 10.3390/cancers14215179

**Published:** 2022-10-22

**Authors:** Vinith Yathindranath, Nura Safa, Babu V. Sajesh, Kelly Schwinghamer, Magimairajan Issai Vanan, Rashid Bux, Daniel S. Sitar, Marshall Pitz, Teruna J. Siahaan, Donald W. Miller

**Affiliations:** 1Department of Pharmacology and Therapeutics, University of Manitoba, Winnipeg, MB R3E 0Z3, Canada; 2Cancer Care Manitoba Research Institute—CCMRI, Winnipeg, MB R3E 0V9, Canada; 3Department of Pharmaceutical Chemistry, The University of Kansas, Lawrence, KS 66047, USA; 4Department of Pediatrics and Child Health, University of Manitoba, Winnipeg, MB R3T 2N2, Canada; 5BioMark Diagnostics Inc., Richmond, BC V6X 2W2, Canada; 6Department of Internal Medicine, University of Manitoba, Winnipeg, MB R3E 0V9, Canada

**Keywords:** glioblastoma (GB), Spermidine/spermine N1-acetyltransferase 1 (*SAT1*), lipid nanoparticles, siRNA, microfluidic mixing, gene therapy, brain drug delivery, blood–brain barrier (BBB), transient modulation, cadherin peptide, tumor sensitization

## Abstract

**Simple Summary:**

Glioblastoma (GB) is an aggressive form of brain cancer with no effective cure. The current treatment for GB involves surgical removal of the tumor followed by chemotherapy and radiation therapy. However, GB develops chemo and radiation therapy resistance, leading to tumor recurrence. GB cells, in comparison to normal cells, have high metabolic rates and adapt several cell signaling pathways to promote their survival. Hence, identifying and inhibiting these tumor-protecting pathways can be helpful in managing GB therapy better. In this study, Spermidine/spermine N1-acetyltransferase 1 (*SAT1*), an enzyme known to cause resistance in GB cells, was targeted and inhibited. Lipid nanoparticles were designed and formulated to target and silence the *SAT1* gene specifically. Inhibiting *SAT1* in GB cells was toxic to the GB cells and further sensitized them towards radiation and chemotherapy.

**Abstract:**

Spermidine/spermine N1-acetyltransferase 1 (*SAT1*) responsible for cell polyamine catabolism is overexpressed in glioblastoma multiforme (GB). Its role in tumor survival and promoting resistance towards radiation therapy has made it an interesting target for therapy. In this study, we prepared a lipid nanoparticle-based siRNA delivery system (LNP-si*SAT1*) to selectively knockdown (KD) *SAT1* enzyme in a human glioblastoma cell line. The LNP-si*SAT1* containing ionizable DODAP lipid was prepared following a microfluidics mixing method and the resulting nanoparticles had a hydrodynamic size of around 80 nm and a neutral surface charge. The LNP-si*SAT1* effectively knocked down the *SAT1* expression in U251, LN229, and 42MGBA GB cells, and other brain-relevant endothelial (hCMEC/D3), astrocyte (HA) and macrophage (ANA-1) cells at the mRNA and protein levels. *SAT1* KD in U251 cells resulted in a 40% loss in cell viability. Furthermore, *SAT1* KD in U251, LN229 and 42MGBA cells sensitized them towards radiation and chemotherapy treatments. In contrast, despite similar *SAT1* KD in other brain-relevant cells no significant effect on cytotoxic response, either alone or in combination, was observed. A major roadblock for brain therapeutics is their ability to cross the highly restrictive blood–brain barrier (BBB) presented by the brain microcapillary endothelial cells. Here, we used the BBB circumventing approach to enhance the delivery of LNP-si*SAT1* across a BBB cell culture model. A cadherin binding peptide (ADTC5) was used to transiently open the BBB tight junctions to promote paracellular diffusion of LNP-si*SAT1*. These results suggest LNP-si*SAT1* may provide a safe and effective method for reducing *SAT1* and sensitizing GB cells to radiation and chemotherapeutic agents.

## 1. Introduction

Glioblastoma (GB) is the most common type of primary brain tumor in adults. As a grade IV astrocytoma, GB is a highly invasive and aggressive form of tumor. With current treatments, including surgical resection of the tumor, radiation and chemotherapy, the median survival time of patients is approximately 16–21 months [1]. While recent advancements, such as immunotherapy and other targeted therapeutics, have dramatically improved outcomes for many cancers, therapeutic advancements for GB have been modest [2]. Improvements in long-term survival (>2 years) within subsets of GB patients suggest there may be molecular differences that can be exploited to achieve better treatment responses [3]. Thus, there is a need to investigate novel strategies to treat GB that utilize the disease’s molecular pathways and genetic heterogeneity in individuals to treat cancer more effectively.

Ongoing cancer molecular biology research has helped identify genes and signalling pathways that promote tumorigenesis or resistance towards conventional therapy [4,5,6,7]. For example, Cullin 1 (*CUL1*), V-Myb avian myeloblastosis viral oncogene homolog-like 2 (*MYBL2*), mutant tumor protein 53 (*TP53*), and hepatoma-derived growth factor (*HDGF*) were identified as tumor promoter genes in GB and used as targets in pre-clinical studies [8,9,10,11]. Recently, Spermidine/spermine N1-acetyltransferase 1 (*SAT1*), an enzyme involved in polyamine catabolism in cells, was identified to promote brain cancer cell migration, proliferation, and resistance towards radiation therapy [12,13]. Furthermore, increased expression of *SAT1* in GB is associated with a decreased overall and progression-free survival (r2l public database). Hence, further studies on the role of *SAT1* as a viable gene target for GB therapy and potential therapeutic approaches for modifying the expression of *SAT1* are of interest.

One approach for modifying select gene targets is RNA interference (RNAi) using small-interfering RNA (siRNA; ~20 base pairs). The siRNA approach uses the cell’s RNAi machinery to form an RNA-induced silencing complex (RISC) that results in the degradation of complementary mRNA and downstream inhibition of protein synthesis. While delivery of the siRNA can be accomplished through both viral and non-viral vectors, the non-viral route has been more extensively explored for clinical applications. Among the non-viral siRNA delivery platforms, lipid-based formulations have been the most successful in crossing over to clinical trials and gaining regulatory approval [14]. The development of microfluidic methods for formulating LNPs has provided distinct advantages in both reproducibility and better control over the physicochemical properties of the LNPs compared to conventional solvent injection methods [15,16]. Cationic polymers such as polyethyleneimine and lipids such as 1,2-dioleoyl-3-trimethylammonium-propane (DOTAP) that complex with anionic siRNA have been successfully used and show promise for targeted gene knock down (KD) in both tumor cell culture and mouse xenograft models [17,18].

Nanotherapeutics for brain diseases have the additional challenge of crossing the blood–brain barrier (BBB) and reaching therapeutically relevant concentrations within the brain. The brain microcapillary endothelial cells (BMEC) present a significant barrier to solutes and macromolecules including anticancer drugs [19]. The BMEC cells form complex tight junctions at the cell–cell interface, presenting a formidable physical barrier for paracellular diffusion. In addition, the transcellular passage of drugs is restricted by various efflux transporters (e.g., Pgp and BCRP) expressed in the BMEC that effectively recycle a wide variety of substrates entering the cells back into the systemic circulation [20]. In general, passive diffusion across the BBB is limited to molecules within a narrowly confined chemical space that has lipophilicity (Log P) between 1–3 and a polar surface area less than 100 Å^2^ [21]. Such restrictions limit the brain penetration of most of the conventional chemotherapeutic agents and virtually all biological and nanomedicine formulations [20]. Hence, the development of brain therapeutics must also include strategies to enhance their delivery across the BBB [20,22,23].

In the present study, we examined the impact of *SAT1* KD on GB survival under both normal conditions and following exposure to radiation and various chemotherapeutics. A biocompatible LNP encapsulating *SAT1*-specific siRNA was identified and used to effectively knock down *SAT1* expression in U251, LN229 and 42MGBA GB cell lines. The ionizing lipid 1,2-dioleoyl-3-dimethylammonium-propane (DODAP) was used as a core component to aid in the electrostatic loading of siRNA while reducing the cell toxicity observed with many cationic lipid formulations. The LNPs were prepared using a microfluidic mixing method to achieve high siRNA encapsulation efficiency and favorable physicochemical properties required for systemic drug delivery applications. The LNP-siRNA formulation was then evaluated for both gene KD efficiency and pharmacological effects, both alone and in combination with ionizing radiation and chemotherapy, in an established GB tumor cell line. Additional proof-of-concept studies were performed using a BBB-GB co-culture model to establish a method for delivery of the LNP to the tumor site using a cadherin binding peptide to modulate brain endothelial cell barrier properties.

## 2. Materials and Methods

1,2-dioleoyl-3-dimethylammonium-propane (DODAP), 1,2-distearoyl-sn-glycero-3-phosphocholine (DSPC), and 1,2-distearoyl-sn-glycero-3-phosphoethanolamine-N-[(polyethylene glycol)-2000] (DSPE-PEG) were obtained from Avanti (Alabaster, AL, USA), cholesterol, *N*-(3-Dimethylaminopropyl)-*N*′-ethylcarbodiimide hydrochloride (EDC) and N-hydroxysuccinimide (NHS) was obtained from Sigma-Aldrich (St. Louis, MO, USA). Benzoxazolium, 3-octadecyl-2-[3-(3-octadecyl-2(3H)-benzoxazolylidene)-1-propenyl]-, perchlorate (DiO) was obtained from Invitrogen (Carlsbad, CA, USA). Recombinant Human Apolipoprotein E (APOE) was purchased from Abcam (Toronto, ON, USA). siRNA targeting *SAT1* gene si*SAT1* (sense 5’→3’ CCAUCCAUCAACUUCUAUAtt and antisense 5’→3’ UAUAGAAGUUGAUGGAUGGtt) and negative siRNA control siSCR were purchased from Ambion by Life Technologies (Carlsbad, CA, USA).

Cell culture: Human GB cell lines U251 (Japanese Collection of Research Bioresources—JCRB; CVCL_0021) and LN229 (American Type Culture Collection—ATCC R CRL-2611, Maryland, USA; CVCL_0393) were grown in Dulbecco’s Modified Eagle Medium: Nutrient Mixture F-12 (DMEM/F12) (Gibco, Carlsbad, CA, USA) with 10% fetal bovine serum (FBS; Gibco), 1% penicillin-streptomycin (Gibco). Human GB cell line 42MGBA (Creative Bioarray, Shirley, NY; CVCL_1798) was grown in 80% mixture of RPMI-1640 (Gibco, Carlsbad, CA, USA) + EMEM (Gibco, Carlsbad, CA, USA) (at 1:1) + 20% FBS. The human brain microvascular endothelial cell line, hCMEC/D3 was obtained from Pierre-Oliver Couraud, INSERM, France (CVCL_U985). The cells (Passage 27–35) were cultured in EBM-2 (Lonza) media supplemented with 5% heat-inactivated FBS (Gibco), 1% penicillin-streptomycin (Gibco Carlsbad, CA, USA), 1.4 μM hydrocortisone (Sigma), 5 μg/mL ascorbic acid (Sigma), 1% lipid concentrate (Invitrogen), 10 mM HEPES (Gibco), and 1 ng/mL basic fibroblast growth factor (Gibco). Primary human astrocytes (HA—CELL Applications, CA, USA) were grown in HA growth medium (CELL Applications, CA, USA) supplemented with 1% FBS and 1% penicillin-streptomycin. The murine macrophage cell line, ANA-1 (gift from Dr. Cynthia Ellison, Department of Laboratory Medicine & Pathobiology, University of Toronto; CVCL_0142), was grown in Dulbecco’s Modified Eagle Medium (DMEM) supplemented with 10% FBS and 1% penicillin-streptomycin. For routine culture, all cells were grown in T75 flasks maintained at 37 °C in a humidified incubator with 5% CO_2_.

Formulation of siRNA encapsulated LNP (LNP-si*SAT1* and LNP-siSCR): Appropriate volumes of lipids from individual stocks were mixed and diluted in ethanol, adhering to the molar ratio of DODAP/DSPC/cholesterol/DSPE-PEG/DiO of 50/10/37.5/1.5/1% and a total lipid concentration of 10 mg/mL. The siRNA (si*SAT1* and control siSCR) was dissolved in sodium acetate buffer (25 mmol, pH = 4) to yield 0.33mg/mL concentration. For the microfluidic mixing, 1 × volume of the lipid organic phase and 3 × volumes of the siRNA aqueous phase were micromixed using a NanoAssemblr Benchtop instrument (Precision NanoSystem, Vancouver, BC, USA). At a flow rate ratio (FRR) of 1:3 (organic:aqueous) and a total flow rate (TFR) of 12 mL/min, the resulting LNP-siRNA had an average diameter of 80 nm. The LNP-siRNA solution was diluted (50-times) in PBS (pH = 7.4) and concentrated using a centrifugal filter (2000× *g*, MWCO 3000) to its original volume. The concentration of the encapsulated siRNA in the LNP-siRNA formulation was measured using a Thermo Scientific NanoDrop spectrophotometer (Madison, WI, USA).

Size and charge determination of LNPs: The particle size (hydrodynamic) and the net surface charge (zeta potential) was measured using ZetaPALS (Brookhaven Instruments, NY, USA) dynamic light scattering instrument. Purified LNP formulations were diluted to 20 µg/mL lipid concentration in PBS (pH 7.4). The polydispersity index (PDI) obtained from the dynamic light scattering instrument was used to determine the size distribution of the LNPs (lower PDI meaning monodisperse LNPs).

Transfection using LNP-si*SAT1*: For transfection, cells were seeded (20,000 cells/cm^2^) in T-25 flasks and grown to 70% confluency. The complete media was replaced with DMEM/F-12 without FBS and antibiotics the day before transfection. LNP-si*SAT1* (7 mL with 1 ug/mL APOE; 80 nM final siRNA concentration in transfection media) was added to the flask and incubated overnight in a CO_2_ incubator at 37 °C. The next day, the treatment was replaced with complete media, and the cells were placed in the CO_2_ incubator. The KD of *SAT1* at the mRNA and protein levels was determined at 48 and 72 h after transfection, respectively, as described below.

Real-Time One-Step RT-PCR: Total mRNA was isolated from cells using TRIzol reagent (Invitrogen, Burlington, ON), following the manufacturer’s protocol. The concentration of isolated mRNA in solution was estimated spectrophotometrically. The one-step qPCR reactions were performed using iTaq^™^ Universal SYBR^®^ Green One-Step Kit (Bio-Rad) following the manufacturer’s protocol. For a 20 µL reaction, 0.5 µg mRNA was used. The primers (Invitrogen) specific for *SAT1* (sense 5’-CTCCGGAAGGACACAGCATT-3’ and antisense, 5’-ACCTCATTGCAACCTGGCTTA-3’) and the internal control 18S (sense 5’-AAACGGCTACCACATCCAAG-3’ and antisense, 5’-CCTCCAATGGATCCTCGTTA-3’) was used. Thermocycling [Reverse transcription: 50 °C (10 min.), Polymerase activation and DNA denaturation: 95 °C (1 min.), 40 cycles (Denaturation: 95 °C (15 sec.), Annealing: 60 °C (60 sec.) and readout)] was carried out using AppliedBiosystems AB7500 instrument. The relative mRNA levels compared to controls were calculated following the 2^−ΔΔCT^ method [24].

Western Blot: Cells were washed with PBS (3×) and lysed using RIPA buffer. Lysates in ice were sonicated for 10 s and centrifuged at 15,000× *g* for fifteen minutes. The supernatant was collected, and the total protein concentration was measured using the Pierce^®^ BCA Protein Assay Kit (Fisher Scientific, Waltham, MA, USA). The lysates (40 µg protein/well) were mixed with 4× loading buffer and separated on 15% polyacrylamide gel or 4–15% precast gels (Min-PROTEAN TGX Stain-Free Gels; Bio-Rad), and subsequently transferred to polyvinylidene difluoride (PVDF) membranes. The PVDF membrane was incubated for an hour in blocking buffer [5% (*w*/*v*) skimmed milk in TBST—tris-buffered saline (pH 7.4) with 0.1% (*v*/*v*) Tween^®^20] for an hour at room temperature. The membrane was incubated overnight with anti-*SAT1* (1/1000 dilution, 10708-1-AP) in 5% (*w*/*v*) skimmed milk in TBST—tris-buffered saline (pH 7.4) at 4 °C overnight. The membrane was washed (4 × 15 min) in TBST buffer and incubated with HRP-conjugated secondary antibody for one hour at room temperature. Later, the membranes were washed (3 × 15 min) in TBST buffer and blots were visualized by enhanced chemiluminescence (BioRad) as per the manufacturer’s protocol.

Cytotoxicity assay: For *SAT1* KD and radiation/anticancer drug combination therapy evaluation, U251, LN229, 42MGBA, hCMEC/D3, ANA-1, HA cells were seeded (10,500 cells/cm^2^) in 24-well plates and transfected following similar conditions as described above. At 72 h post-transfection, the cells were exposed either to radiation (1, 3, 6, 10 or 15 Gy; RS-2000, Rad Source Technologies, Inc., Buford, GA, USA) or various anticancer drugs, BCNU (100 µM), Doxorubicin (0.1 µM) and Topotecan (0.2 µM). After 24 h, the media and treatments were removed and replaced with fresh media and cultured for 48 h. The percentage of viable cells was determined using the MTT assay [25].

Synergistic activity was evaluated using a coefficient of therapy interaction (CTI). The CTI values were calculated using the following equation: CTI = AB/(A × B), where AB is the ratio of the absorbance (MTT A570) of the *SAT1* KD combined with radiation therapy to the control (media), and A and B are the ratios of the individual treatment groups (*SAT1* KD and radiation therapy) to the control [26,27]. A CTI < 1 indicates synergism, while a CTI = 1 indicates an additive response and CTI > 1 indicates antagonism of the combined therapies.

Comet assay: The comet assay was performed using a kit (Abcam) following the manufacturer’s protocol. The assays on control, SAT KD and irradiated U251, LN229 and 42MGBA cells were performed at 6 and 24 h after radiation exposure. The comet assay involves single-cell DNA gel electrophoresis, where DNA damage is quantified based on DNA in the nucleus (comet head) and damaged fragments that travel across the gel (Tail). The extent tail moment (Tail DNA% × Length of Tail) calculated gives insight into the extent of DNA damage. The extent tail moment was analyzed using the OpenComet plugin for ImageJ. At least 50 individual cells per treatment were analyzed.

γ-H2AX immunofluorescence: Phosphorylated Serine (139) on histone variant H2AX were detected using antibodies at a dilution of 1:1000 (Cell signalling, 2577S) as described previously with minor modifications [28]. U251 cells were treated with LNP-si*SAT1* (80 nM) and, after 48 h, were seeded on coverslips (20,000 cells/cm^2^). After 24 h, the cells were exposed to 1 Gy radiation and incubated for 6 h. Cells were fixed with freshly prepared 4% Paraformaldehyde (Fisher Scientific, Waltham, MA, USA) in phosphate-buffered saline (PBS; 0.01M; pH 7.4) for 20 min. Following fixation, cells were washed with PBS, permeabilized with 0.5% Triton-X-100 in PBS and incubated with the primary antibody for one hour at room temperature. Cells were washed with 0.1% Triton-X100 (in PBS) followed by PBS and incubated with the secondary antibody (AlexaFluor™488; 1:200; Thermofisher) for one hour at room temperature. Cells were washed with 0.1% Triton-X100 (in PBS) followed by PBS, and nuclei were counterstained with DAPI (Abcam; ab104139). Microscopy was performed as described with minor modifications [29]. Briefly, 30 cells were imaged on an AxioImager 2 (Zeiss) equipped with an AxioCam HR charge-coupled device (CCD) camera (Zeiss) and a 63× oil immersion plan-apochromat lens (1.4 numerical aperture). Images were acquired with AxioVision software and saved as 16-bit Tiff images. Nuclei were imaged on the blue channel (pseudo-colored red for illustration purposes), and γ-H2AX foci were imaged on the green channel (pseudo-colored green for illustration purposes). γ-H2AX foci were enumerated using ImageJ using the plugins as described elsewhere (https://microscopy.duke.edu/guides/count-nuclear-foci-ImageJ; accessed on 1 June 2021) data were exported into Prism for statistical analysis. Images were processed using Imaris cell imaging software (Oxford Instruments) to separate channels, and pseudo color nuclei and γ-H2AX. Image panels were generated using Photoshop CS5 (Adobe).

BBB-GB co-culture model: a BBB-GB co-culture model was used to assess the delivery of the LNP-si*SAT1* formulation across brain microvessel endothelial cells [25]. For these studies, human immortalized brain endothelial cells (hCMEC/D3) were grown to confluency on Transwell inserts (0.4-micron pore size), and after reaching confluency, the inserts were placed in 12-well plates containing U251 tumor cells (approximately 70% confluency). For the transfection/permeability studies, media was removed from the top (donor compartment) and bottom (receiver compartment) of the inserts and replaced with transfection media (hCMEC/D3 media described above without FBS and antibiotics). Following a 30 min pretreatment with either PBS (control) or cadherin binding peptide (ADTC5; 1 mM), the LNP-si*SAT1* (80 nM) was added to the donor compartment (1.5 mL) along with APOE (1 µg/mL) and incubated for two hours at 37 °C with shaking (50 RPM). After two hours, the donor compartment with the hCMEC/D3 monolayer was removed. The U251 cells in the receiver compartment were further incubated for 6 h in the humidified CO_2_ incubator to aid transfection. After six hours, the media was changed to complete U251 media and *SAT1* KD was determined at 48 h as described above. To assess the modulatory effects of the cadherin peptide on monolayer permeability, a fluorescent paracellular permeability marker IRdye800-PEG (0.1 µM) was added to the donor compartment at the start of the transfection treatment. The concentration of IRdye 800-PEG in the donor compartment and the receiver compartments were measured fluorometrically (Ex: 750 nm and Em: 782 nm). Permeability was expressed as the percent flux determined by dividing the cumulative concentration of the dye in the receiver compartment (t = 2 h) by the concentration in the donor compartment (t = 0).

Statistical analysis: All data are expressed as the mean ± standard error of the mean (SEM). Statistical analysis was performed using one-way or two-way ANOVA, followed by Tukey’s test. In all studies, *p* < 0.05 was considered statistically significant.

## 3. Results

The si*SAT1* encapsulated LNP (DODAP/DSPC/cholesterol/DiO/DSPE-PEG) was formulated following the microfluidic mixing procedure outlined in Figure 1. The hydrodynamic size of the LNP-si*SAT1* was estimated to be 82 nm with a net neutral surface charge (zeta potential: 0.18 ± 0.42). The particles displayed a polydispersity index (PDI) of 0.16, indicating the monodisperse nature of the nanoparticles. The UV spectroscopic (A260/A280) analysis of siRNA in both LNP-si*SAT1* and filtrate showed a high encapsulation efficiency of 100%. The LNP-siRNA formed a highly stable but hazy dispersion in PBS (pH 7.4) and was stored at 4 °C and −80 °C for the short and long term, respectively, without significant loss of stability or biological activity.

As a part of LNP-si*SAT1* development trials, different formulation (N/P ratio—amine in DODAP/siRNA phosphate) and transfection (siRNA concentration and time) parameters were investigated. Initially, LNP-si*SAT1* with N/P ratios of 5, 10 and 15 were prepared. The ability of LNP-si*SAT1* to deliver siRNA and knock down the target *SAT1* mRNA was evaluated in U251 cells (Appendix A). The cells were transfected with LNP-si*SAT1* in the presence of APOE (1 µg/mL). The LNP-si*SAT1* with N/P ratio of 5 and 10 did not produce measurable *SAT1* KD even at the higher 80 nM si*SAT1* concentration (Appendix A). However, the LNP-si*SAT1* with an N/P ratio of 15 produced a significant KD at 40 nM (~70%) and 80 nM (~80%) concentrations (Appendix A). The LNP-si*SAT1* at N/P ratio of 15 in the presence of APOE (1 µg/mL) produced the maximum *SAT1* KD at the mRNA level and was used for further studies (Figure 2A,B). The Western blot (WB, Figure 2C) and *SAT1* immunofluorescence (Figure 2D,E) studies showed a reduction in *SAT1* in the LNP-si*SAT1* transfected cells compared to cells treated with scrambled siRNA control. Transfection of LN229 and 42MGBA cells using LNP-si*SAT1* produced around 80% SAT1 KD at the mRNA level and around 50% at the protein levels, respectively (Figure 2F,G). We also investigated cationic LNP-si*SAT1* containing DOTAP, which carries a permanent cationic charge. The cationic LNP-si*SAT1* at N/P ratio of 5 and siRNA concentration of 40 nM produced 80% *SAT1* KD at the mRNA level. However, the cationic LNP (based on LNP-siSCR) was toxic, resulting in around a 30% drop in cell viability (based on MTT studies) and was not used for further studies.

The effects of *SAT1* KD on U251 cell viability under control conditions and following exposure to radiation and anticancer drugs doxorubicin, BCNU and topotecan, which are known to cause DNA double-strand breaks, was examined [30,31,32]. Cell viability after *SAT1* KD and following radiation or chemotherapeutic exposure was assessed using the MTT assay (Figure 3). Knocking down *SAT1* expression using LNP-si*SAT1* in U251 cells resulted in a 40% reduction in cell viability (Figure 3A). Despite similar levels of reduction in *SAT1* expression (Appendix A), no decrease in cell viability was observed in the other cells examined, including microvascular endothelial (hCMEC/D3), macrophage (ANA-1) and astrocyte (HA) cells, following treatment with LNP-si*SAT1*(Figure 3B–D). The cytotoxic response to various chemotherapeutic agents was modestly increased following *SAT1* KD with LNP-si*SAT1* in U251 cells (Figure 3A). However, the most dramatic increase in cytotoxic response was observed following radiation treatment in the *SAT1* KD group, where LNP-si*SAT1,* when combined with radiation treatment, resulted in approximately 15% cell viability compared to approximately 45% cell viability in the scrambled siRNA control group following radiation exposure (Figure 3A). Such sensitization to *SAT1* KD combination therapy was absent in the hCMEC/D3, HA and ANA-1 cells (Figure 3). In the case of LN229 and 42MGBA GB cell lines, the knock-down of *SAT1* did not result in any observable loss of cell viability. However, the *SAT1* KD LN229 and 42MGBA cells, when exposed to radiation, displayed cell viabilities of ~40% and ~44%, respectively, which was significantly lower than the irradiated control groups (LN229—74% and 42MGBA 65%) (Figure 3E,F). The nature of sensitization in GB cells with *SAT1* KD and radiation/anticancer drug combination therapy was further evaluated using the Coefficient of Therapy Interaction (CTI). The combination group with *SAT1* KD and radiation displayed a CTI value <0.7 in all the ranges of radiation doses studied (1, 3, 6 and 10 Gy) (Figure 4). All the *SAT1* KD and anticancer combination groups had a CTI value >0.7.

To further understand the cellular mechanisms responsible for the effects of *SAT1* KD in the U251, LN229 and 42MGBA cells, the extent of DNA damage in control and *SAT1* KD cells post-irradiation was examined using a comet assay (Figure 5). Here, the *SAT1* KD and control cells were exposed to radiation (U251—10 Gy; LN229 and 42MGBA—15 Gy), and the Comet assay was performed after six hours (Figure 5). The calculated extent tail moments (Figure 5C,M,P) in *SAT1* KD cells (Figure 5B,L,O) were found to be 1.5-fold higher than in the control cells exposed to radiation, indicating a higher amount of DNA damage in irradiated U251, LN229 and 42MGBA cells with reduced *SAT1* expression (Figure 5). It is worth mentioning that *SAT1* KD by itself did not display DNA damage as observed using comet assay (Appendix A). γ-H2AX is a well-established marker for DNA double-strand breaks and repair [33]. An early cellular response to DNA double-strand breaks is the rapid phosphorylation of H2AX at ser139 to form γ-H2AX. Thus, γ-H2AX levels are directly correlated with the extent of DNA double-strand breaks. Immunofluorescence on U251 cells exposed to 1 Gy and 10 Gy radiation displayed γ-H2AX foci in the nucleus, while the higher 10 Gy dose displayed a considerably higher number of foci, indicating higher instances of DNA DSB (Figure 5E,F,H,I and Appendix A). In U251 cells treated with LNP-si*SAT1* and exposed to 1 Gy radiation (Figure 5I), there was a higher (4-fold; Figure 5J) number of γ-H2AX foci detected after six hours compared to cells receiving only radiation, indicative of the presence of a higher number of DNA DSB in *SAT1* KD U251 cells [33]. Together, these studies suggest that *SAT1* expression is correlated with observed DNA damage in U251 cells exposed to radiation.

As a proof of concept, the ability of LNP-si*SAT1* to deliver siRNA across hCMEC/D3 monolayer co-cultured with U251 cells was evaluated (Figure 6). The integrity of the hCMEC/D3 monolayer was tracked based on the percent flux of a large molecular weight IRdye-PEG (35,000 Da) permeability marker. The cadherin binding peptide ADTC5 treated hCMEC/D3 monolayers displayed a 2.8-fold higher flux of IRdye-PEG compared to the control cells. The higher flux of the macromolecule permeability marker indicated that the ADTC5 disruption was successful, thereby enabling the paracellular diffusion of the hydrophilic dye. The permeability of LNP-si*SAT1* across the hCMEC/D3 was evaluated based on transfection and *SAT1* KD in U251 co-culture (Figure 6C). In the group with ADTC5 disruption, a significant mRNA KD of ~37% was observed. In the group without ADTC5 to modulate barrier integrity, no significant *SAT1* KD was detected. Based on these preliminary studies, the LNP-si*SAT1* formulation combined with the ADTC5 cadherin peptide to modulate brain microvessel endothelial cell permeability may be an effective method for delivering siRNA to GB targets in the brain.

## 4. Discussion

The present study examined a siRNA-based approach targeting *SAT1* expression in GB. While siRNA-based therapeutics have enormous potential for targeted silencing of critical pathways involved in cell signaling and metabolism in cancer [34,35], safe and effective delivery is crucial. In this regard, LNPs have both biocompatibility and loading efficiency as merits for consideration. The lipid composition of the LNPs plays a significant role in determining the drug entrapment efficiency, size, surface charge and blood circulation half-life. Usually, a mixture of lipids is used in LNP formulation to achieve the desired physicochemical and drug loading/delivery properties. We focused on DODAP, DSPC, cholesterol, DSPE-PEG2000 and DiO as the lipid components for our initial formulation trial. DODAP is an ionizing lipid (PKa: 6.6–7) that carries a cationic charge at acidic pH and, therefore, can electrostatically bind to the negatively charged siRNA. Hence, the high encapsulation efficiency (>95%) of siRNA observed in the present study could be attributed to the presence of DODAP in the lipid mix. The LNPs used in the present study also included a PEG lipid component to aid in the optimization of both the size and stability of the LNPs. It has been reported that increasing the PEG lipid composition can reduce the size of the LNPs [36]. While highly hydrophilic nanoparticles smaller than 10 nm in diameter are rapidly cleared from the circulation via extravasation and renal clearance [37], nanoparticles larger than 200 nm in diameter are efficiently removed by the reticuloendothelial system [38]. As our intended target resides within the brain, we postulated that LNP sizes between 60–100 nm in diameter would have the greatest chance for delivery across the BBB when combined with the transient opening of the BBB using cadherin peptides. The incorporation of PEG (1.5%) into the LNP formulation resulted in a LNP of approximately 80 nm in diameter that retained a high siRNA encapsulation efficiency. While nanoparticles with a net positive surface charge have better cellular uptake than the neutral and negative surface-charged counterparts, cationic nanoparticles suffer from rapid clearance by nonspecific binding and phagocytosis [39]. The LNP-siRNA used in the present study has physicochemical properties that are likely to provide for long circulation half-life and have a better chance of accumulating at the tumor site by extravasation or crossing a transiently opened BBB.

Though surface PEG is beneficial for LNP formation, stability, and circulation half-life, it can have a negative impact on transfection efficiency. Having a higher amount of PEG-lipid on the surface was shown to adversely affect the cell uptake and endosomal escape of LNPs [40]. To overcome the limitations of PEGylation on cellular LNP uptake, APOE was added to the LNP formulation. The presence of APOE has been reported previously to facilitate LNP endocytosis through an APOE-dependent low-density lipoprotein receptor (LDLR) pathway [41] and provided for improved knockdown efficiency with the LNPs in the current study.

Polyamines such as spermidine and spermine play essential roles in cell functions, including maintaining chromatin structure, facilitating growth, proliferation of the cell, regulating ion channels, and scavenging free radicals [42,43]. Intracellular polyamine levels in normal cells are tightly controlled by biosynthesis, catabolism and transport systems. Polyamine levels in several cancers, including GB, are dysregulated and exploit their metabolism pathway for survival and proliferation [43,44,45,46]. Hence, early therapies looked at polyamine pathway inhibitors such as DFMO, MGBG, SAM486A, etc. [47]. However, chemotherapy with these inhibitors was largely ineffective in clinical trials. As the polyamine pathway is an essential player in tumorigenesis, it is still of considerable interest as a therapeutic target. In this regard, RNAi and gene therapy holds great promise for targeting and inhibiting specific enzymes involved in polyamine metabolism and warrants further investigation. A key player in polyamine metabolism is *SAT1.* This highly inducible enzyme is responsible for acetylating polyamines which in turn are rapidly excreted from the cells. In gliomas, higher polyamines and activation of polyamine metabolism have been associated with cell proliferation [43,46]. Studies suggest that *SAT1* likely has a protective effect on cancer cells with abnormally high polyamine levels [48]. Excess polyamines in cancer cells are acetylated by *SAT1* for transport and recycling. As excess amounts of polyamines are toxic to the cells, the *SAT1* system is important for conversion to less toxic acetylated forms. We hypothesize that the lower cell viability observed in *SAT1* KD U251 cells in the present study may be attributed to reduced polyamine metabolism [48]. Both LN229 and 42MGBA cells had a doubling time of around 32 h in culture. This proliferation rate was considerably longer than U251 cells with a doubling time of 6 h. Thus, the higher proliferation rate in U251 cells could contribute to the loss of cell viability observed with *SAT1* KD alone. An additional consideration is the baseline expression levels of SAT1, which tended to be greater in the U251 compared to other GB cell lines.

In addition to the direct effects of *SAT1* KD on cell proliferation, a significant enhancement in sensitivity towards radiation and chemotherapy was observed in U251 cells. Previous studies have proposed that *SAT1* may have an important role in DNA damage repair pathways [5]. Previous studies have shown *SAT1* promotes DNA double-strand break repair by regulating *BRCA1* and homologous recombination, aiding in tumor survival [12]. Ionizing radiation causes cell death by DNA damage, specifically double-strand breaks (DSB). In the present study, significant increases in DSB were observed when radiation exposure was combined with LNP-si*SAT1* treatment. Increases in DNA damage observed with combined radiation and *SAT1* KD could reflect either increased production or reduced repair of DSB. However, given the previous findings and the absence of any direct DNA damaging effects from the LNP-si*SAT1* treatment itself, the sensitization observed in U251 cells following *SAT1* KD and radiation treatment in the present study most likely reflects impaired DNA DSB repair. The potential for *SAT1* KD in combination with radiation and chemotherapy to lower the cumulative dose requirement and improve the efficacy and safety profile of existing treatment regimens is encouraging. Given the adverse effects of chemotherapy and radiation therapy, both acute and chronic, combination treatment approaches reducing dose intensity and producing the same, or greater therapeutic effect are attractive [49,50].

Mechanistically, *SAT1* could impact DSB repair through altering polyamine levels within the cell [13]. Alternatively, previous studies have suggested *SAT1* may act independently of its enzymatic effects on polyamines as an important transcription factor for genes that regulate DNA repair [13]. If the radiation sensitization effects are due to *SAT1* acting in a transcriptional manner, then decreasing *SAT1* expression through siRNA-based therapeutics may be more effective than small molecule inhibitors of the enzyme that block activity but do not affect SAT1 expression.

While *SAT1* could be an attractive therapeutic target for GB, delivery of the siRNA across the BBB is a significant challenge. Nanotherapeutics like LNP-si*SAT1* in circulation are unlikely to cross the BBB due to their large sizes. In high-grade tumors like GB, the vasculature forms the brain-tumour barrier (BTB) which may be leakier than the BBB [51]. However, even within the BTB, there is heterogeneity with some vascular regions having intact tight junctions and active efflux transport that would limit drug delivery to the tumor site [52]. Additionally, a progressive glioblastoma can still have an intact BBB. Hence, for any brain tumor therapeutics, it is vital to plan suitable design and delivery strategies at the early stages of development. Enhanced drug delivery to the brain has been achieved by three broad methods: (1) bypassing the BBB entirely by direct administration into the brain, (2) modifying the therapeutic molecule to enhance stability and permeability through passive diffusion or carrier and receptor-mediated routes, and (3) transiently disrupting the tight junction complexes for enhanced paracellular entry [53]. Intracerebroventricular and intracerebral administration (with or without convection-enhanced diffusion) involving direct administration of drugs to the brain has been examined, as has local brain drug delivery approaches using surgical implants made of biodegradable materials (e.g., hydrogel, Gliadel) [54,55,56,57]. Clinical improvements with these approaches have been limited by the poor diffusion distance of therapeutics within the intercellular matrix of the brain parenchyma and potential damage to brain structures.

Systemic drug delivery routes across the BBB is also challenging, but it offers a vast surface area (15–20 m^2^) and short diffusion distances (<25 mm) to any part of the brain parenchyma [58]. Chemical modification of therapeutics to target receptors expressed in the BBB has been widely investigated for cell-mediated endocytosis. In the case of nanoparticles, conjugation to transferrin and EGF has been widely used for receptor-mediated endocytosis across the BBB [59,60]. Gonzales-Carter et al. used a novel two-step targeting for delivering nanomicelles across the BBB in cell culture and mice studies [61]. First, the brain capillary endothelial cells were biotinylated using biotin-α-PECAM1 antibody. The biotin label was used to anchor streptavidin labelled micelles facilitating adsorptive mediated transcytosis across the BBB. Dan et al. used α-PECAM1 antibody-modified iron oxide nanoparticles to enhance the permeability across hCMEC/D3 and rat brain [62]. It is worth noting that while increased nanoparticle delivery to the brain was observed, even with these targeting strategies, only a small percentage of the injected dose actually reached the brain following cell-mediated endocytosis.

Strategies such as osmotic (hypertonic mannitol) and pharmacological (Bradykynin Analogs, alkylglycerols, lysophosphatidic acid and cadherin binding peptides) disruption of the BBB have been reported to enhance paracellular permeability and the delivery of drugs to the brain [23,25,63,64]. As a major advantage, the transient disruption method does not require specific modifications to the therapeutic agents to achieve delivery across the BBB. Of all the delivery methods, osmotic disruption using mannitol has been studied extensively and has been used successfully in the clinic to improve the delivery of drugs to the brain tumor [65,66]. A major drawback with this classical disruptor is the prolonged duration of the BBB opening and the difficulty controlling the magnitude of BBB disruption, both of which can contribute to toxicity associated with the prolonged influx of small and large circulating molecules [67,68].

The cadherin binding peptides were developed as a pharmacological tool for modulating BBB permeability via a short, reversible opening of the intercellular junctions controlling paracellular diffusion of solutes [23]. The cadherin peptides, by design, bind to the EC domain of E-Cadherin, a membrane protein of the *adherens* junction of the BBB [69]. The peptide-E-cadherin binding inhibits the cadherin-cadherin homodimer interactions between adjacent brain capillary endothelial cells resulting in the disruption of the BBB tight junction. Previous studies have identified ADTC5 peptide as being capable of modulating BBB permeability to both small and select large macromolecules with rapid restoration of BBB integrity [23,25,63,64,70]. Particularly relevant for the current studies was the ability of ADTC5 to increase the penetration of iron oxide nanoparticles in an in vitro BBB model [25]. In the current study, the cyclic cadherin binding peptide (ADTC5) was used as a BBB permeability enhancer [22,71]. In this proof-of-concept study, we have demonstrated that transient opening of the BBB using cadherin peptides can enhance LNP delivery across the BBB. This could potentially be extended to other LNP based drug delivery carriers for the brain.

## 5. Conclusions

In summary, the LNP-si*SAT1* formulation displayed a high siRNA encapsulation efficiency, low polydispersity index and neutral surface charge. The LNP-si*SAT1* effectively delivered si*SAT1* in the GB cell lines producing significant KD of *SAT1* at both the messenger and protein levels. Combining the LNP-si*SAT1* with cadherin binding peptide could potentially enhance the delivery of siRNA therapeutics across the BBB. Reduced *SAT1* enzyme levels adversely affected the U251 viability and sensitized them towards radiation and chemotherapy. Importantly, *SAT1* KD did not produce a similar effect in brain microvascular endothelial, astrocyte and macrophage cell lines. While additional translational studies with mouse xenograft models are required, these initial findings suggest that modulation of *SAT1* beneficially impacts tumor viability both alone and in combination with radiation or chemotherapy and provides a conceptual framework for a nanomedicine-based approach for the delivery siRNA to the tumor site. The ability of *SAT1* gene knockdown to increase DNA-double strand breaks in response to radiation exposure is encouraging for the potential use of si*SAT1*-LNP as a combination therapy to improve the therapeutic effect and lower the cumulative dose requirements for radiation and chemotherapy in patients.

## Figures and Tables

**Figure 1 cancers-14-05179-f001:**
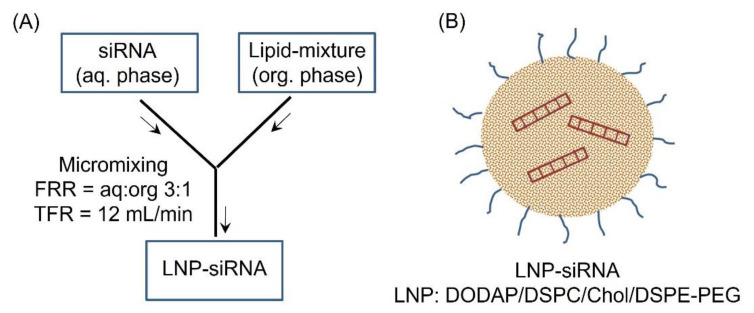
(**A**) Schematic illustration of the formulation of LNP-siRNA following micromixing in a microfluidic chamber. The flow rate ratio (FRR) of aq:org of 1:3 and a total flow rate of 12 mL/min yielded siRNA encapsulated LNP (**B**) that were in the 80 nm (PDI = 0.16) size range with neutral surface charge.

**Figure 2 cancers-14-05179-f002:**
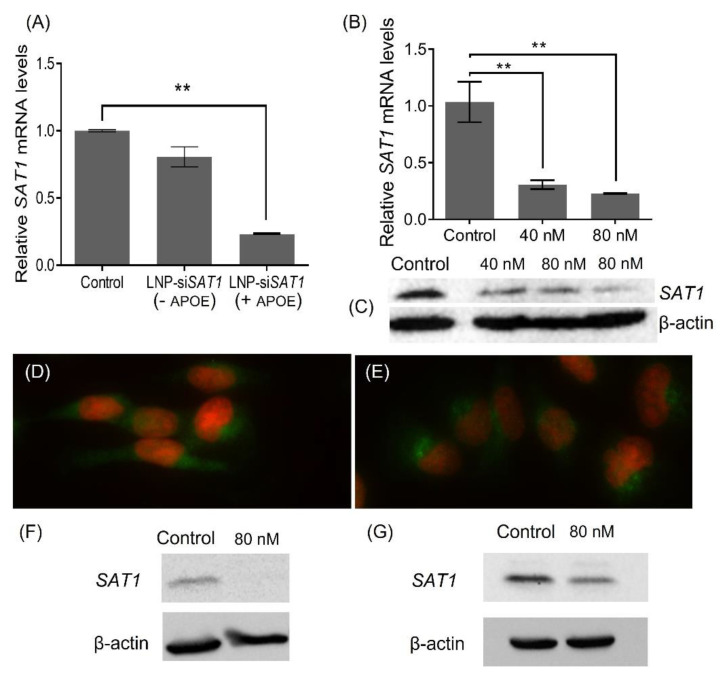
Relative *SAT1* mRNA levels in LNP-si*SAT1* transfected U251 cells against control which received LNP-siSCR, after 72 h: (**A**) with and without APOE (1 µg/mL) in media and (**B**) at 40 and 80 nM siRNA concentration (n = 3, *p* < 0.01). *SAT1* protein knockdown measured using WB (**C**) after 72 h of transfection with LNP-si*SAT1*. β-actin was used as the internal control. Immunofluorescence of *SAT1* (green) and DAPI stain for the nucleus (red) in the control (**D**) and *SAT1* KD U251 (**E**) cells after 72 h. Relative *SAT1* protein levels in LNP-si*SAT1* transfected 42MGBA (**F**) and LN229 (**G**) cells against control which received LNP-siSCR, after 72 h: Values are expressed as the mean ± standard error of the mean (SEM). ** *p* < 0.01 (*p* < 0.05 was considered statistically significant). The uncropped blots are shown in Appendix A.

**Figure 3 cancers-14-05179-f003:**
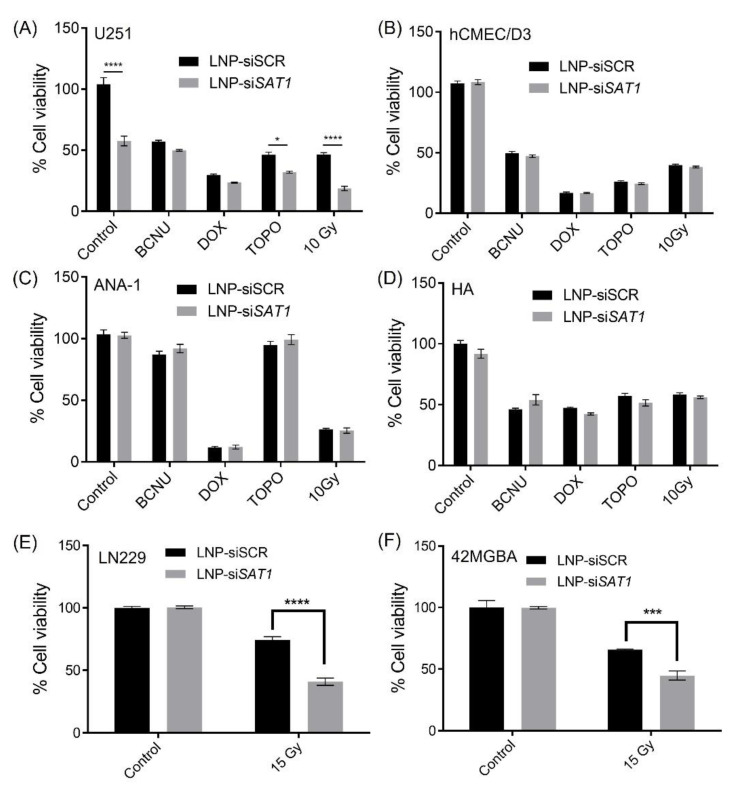
Cytotoxic response to chemotherapy and radiation in various cells following exposure to LNP with scrambled siRNA (control) or *SAT1* siRNA (LNP_si*SAT1*). (**A**) U251, (**B**) hCMEC/D3, (**C**) ANA-1, (**D**) HA cells exposed to chemotherapeutic agents BCNU (100 μM), Doxorubicin (0.1 μM) and Topotecan (0.2 μM), and radiation (10 Gy); (**E**) LN229 and (**F**) 42MGBA cells exposed to 15 Gy radiation. The cytotoxic response is expressed as a percentage compared to cells receiving no chemotherapeutic agents or radiation exposure. Values are expressed as the mean ± standard error of the mean (SEM). Statistical analysis was performed using one-way or two-way ANOVA, followed by Tukey’s test (*n* = 3). * *p* < 0.05, *** *p*<0.001, **** *p* < 0.0001 (*p* < 0.05 was considered statistically significant).

**Figure 4 cancers-14-05179-f004:**
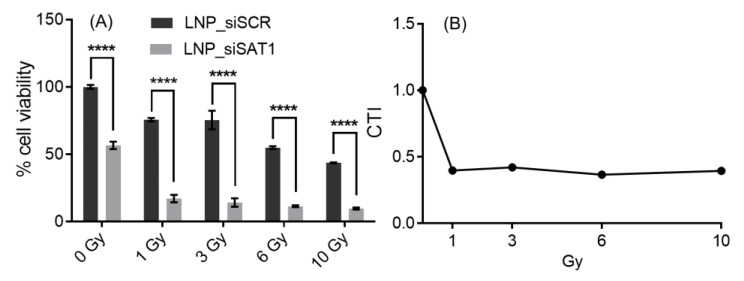
Cytotoxic response to different doses of radiation in *SAT1* KD U251 cells (**A**). The corresponding calculated CTI values (**B**) for plot A values; CTI < 0.7 indicates a significant synergistic effect. Values are expressed as the mean ± standard error of the mean (SEM, *n* = 3). Statistical analysis was performed using two-way ANOVA, followed by Tukey’s test (*n* = 3). **** *p* < 0.0001 (*p* < 0.05 was considered statistically significant).

**Figure 5 cancers-14-05179-f005:**
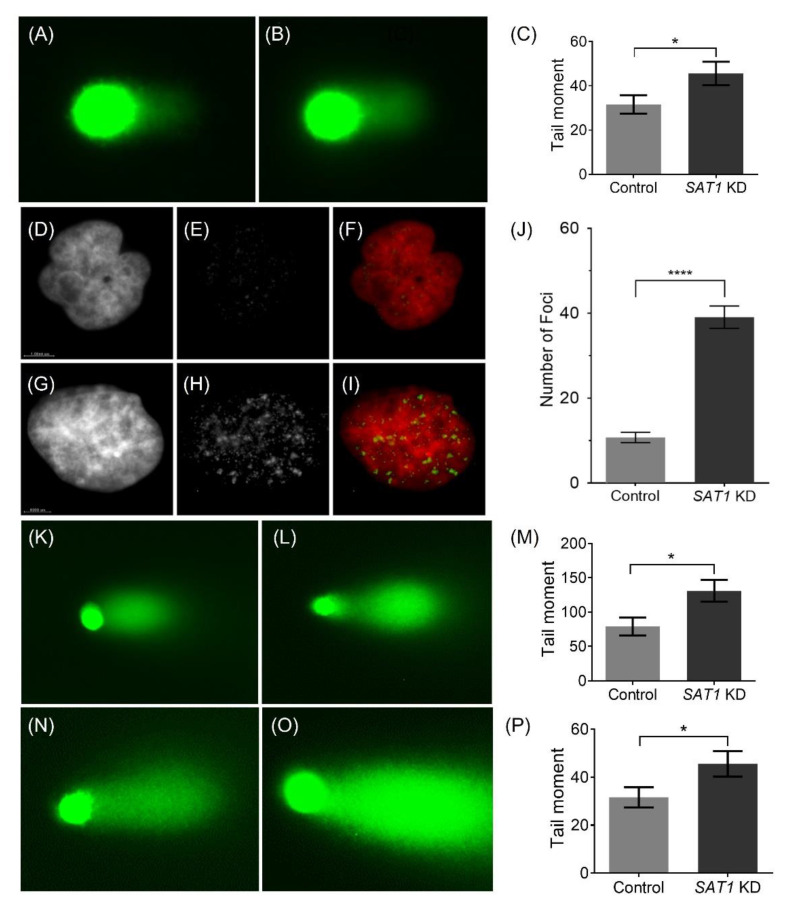
Representative single-cell alkaline gel electrophoresis (Comet Assay) of control (**A**) and *SAT1* KD U251 cells (**B**) 6 h after irradiation (10 Gy) (**C**). Representative immunofluorescence images of γ-H2AX in control (**D**–**F**) and *SAT1* KD (**G**–**I**) U251 cells, 6 h after irradiation (1 Gy). Images of the nuclei stained with DAPI (D—control; G—*SAT1* KD) and γ-H2AX (E—control; H—*SAT1* KD). Images F (D and E overlay) and I (G and H overlay) are overlayed images pseudo colored for nuclei (red) and γ-H2AX (green). Quantitative assessment of γ-H2AX foci based on the average counts from a minimum of 30 individual cells (**J**). Representative single-cell alkaline gel electrophoresis (Comet Assay) of LN229 (control—(**K**), *SAT1* KD—(**L**), extent tail moment—(**M**)) and 42MGBA (control—(**N**), *SAT1* KD—(**O**), extent tail moment—(**P**)) cells, six hours after irradiation (15 Gy). The cells in comet assay were stained with SYBR green and observed under a fluorescence microscope. The Extent Tail Moment (Tail DNA% × Length of tail) was calculated from fifty individual cells. Values are expressed as the mean ± standard error of the mean (SEM). Statistical analysis was performed using unpaired *t*-test (*n* = 3). * *p* < 0.05, **** *p* < 0.0001 (*p* < 0.05 was considered statistically significant).

**Figure 6 cancers-14-05179-f006:**
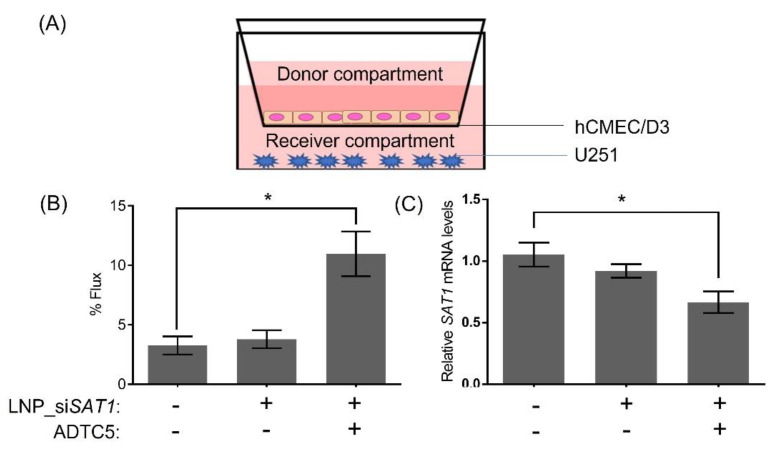
Schematic illustration (**A**) of in vitro BBB co-culture model showing the orientation of the BBB (hCMEC/D3) and glioblastoma (U251) cells. (**B**) 2h flux of the large molecular weight permeability marker IR-dye PEG across non-treated and ADTC5 and LNP-si*SAT1* (in the donor compartment) treated hCMEC/D3 monolayers. (**C**) Changes in *SAT1* mRNA levels in U251 cells from co-culture model following treatment with LNP-si*SAT1* with and without ADTC5, the permeability enhancing peptide. The expression of *SAT1* was determined 48 h following treatment. Values are expressed as the mean ± standard error of the mean (SEM). Statistical analysis was performed using one-way ANOVA, followed by Tukey’s test (*n* = 3). * *p* < 0.05 (*p* < 0.05 was considered statistically significant).

## Data Availability

The datasets used and/or analyzed during the current study are available from the corresponding author on reasonable request.

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
