# Peer review of "Spermidine/Spermine N1-Acetyltransferase 1 (SAT1)—A Potential Gene Target for Selective Sensitization of Glioblastoma Cells Using an Ionizable Lipid Nanoparticle to Deliver siRNA"

_cancers, 2022, doi:10.3390/cancers14215179_

Round 1

Reviewer 1 Report

For the most part, this is a sound study. However, there are some issues that require clarification and a few oversights that I listed below:

Lines 53-58: While the authors’ main point here is well-taken, it is not true that nothing changed since Stupp et al. 2005. Two-year-survival, e.g., has improved substantially in recent years with new treatment strategies, at least in some subpopulations (e.g., Stupp et al. 2017 / TTF, or Herrlinger et al. 2019 / CeTeG trial).

Line 62: heparin-binding growth factor…? “HDGF” = hepatoma-derived growth factor

Line 67: “GMB” should be GBM

Line 287: I assume you refer to Figure 2A (not Figure 1A) here..?

Line 287: Abbreviation “WB” needs to be introduced

Fig 2:

Line 296: “fold change” – what does that mean?

Line 296/297: “..after 72: A)..” – after 72 what?

Line 298: “..knockdown with measured after 72 hours..” – with what?

Figure Legend: Explain for what purpose beta-actin was included (2 C), make clear what red fluorescence is (C) and D)), and clarify what part of the Figure Legend for Fig. 2 refers to C) and D)

Fig 3:

Please specify the number of experiments underlying all your cell viability analyses / the actual “n” underlying your statistical analyses.

Explain how cell viability can be higher than 100% (LNP-siSCR in A), B), and C), and LNP-siSAT1 in B)).

Is there a reason why the order of agents is different in B) compared to A), C) and D)? If not, all plots should be in the same order.

I suggest to label all your plots (A-D) with the name of the cell line it represents to improve readability.

Fig. 4:

Please specify the number of experiments underlying all your cell viability analyses / the actual “n” underlying your statistical analyses.

Fig 5:

Line 350: should be “fluorescence microscope”

Line 353: “images E and F are the corresponding nuclei..” – I suppose you mean E and H

Since this is already a color figure, I would strongly recommend to show D & G in green and E & H in red. This would immediately make clear that both F and I are overlays.

Fig. 6:

Please specify the number of experiments underlying all your flux / permeability studies (B) and mRNA level measurements (C) so that the actual “n” underlying your statistical analyses is revealed.

Line 512: “…that SAT1 is a key factor… in upstream DNA damage repair, and could be targeted using..”

Discussion:

There should be a thorough, explicit discussion of the study’s limitations, such as the cell line used, the (tumor) cell viability model, the blood brain barrier model etc.

If your findings are supposed to lay the groundwork for actual treatments, i.e., if they are supposed to be relevant, what are the next steps? What challenges need to be overcome until there can be clinical trials in humans?

Author Response

We thank all the reviewers for their valuable considerations and suggestions, which helped us improve the paper. Please find below our answers to all comments and concerns, point by point. All changes in the manuscript are highlighted in red to allow easy identification.

Reviewer 1:

Lines 53-58: While the authors’ main point here is well-taken, it is not true that nothing changed since Stupp et al. 2005. Two-year-survival, e.g., has improved substantially in recent years with new treatment strategies, at least in some subpopulations (e.g., Stupp et al. 2017 / TTF, or Herrlinger et al. 2019 / CeTeG trial).

Our intent in this paragraph was to highlight that despite some pretty dramatic improvements in overall survival of patients with many different types of cancer, the prognosis of patients with GB remains pretty poor with median survival rates of around 16 months from first diagnosis. The reviewer is correct that use of tumor treatment fields and standard temozolomide treatment as reported by Stupp et al in 2017 was able to increase overall survival from 16 to 21 months. The tumor treatment field applies a low level magnetic current to discrete sites in the brain and in combination with chemotherapy did yield a statistical improvement. As highlighted in this same paper these findings are in noticeable contrast to the more than 20 + clinical trials in GB using various new agents or higher intensity doses of conventional therapy that failed to show any improvement. While Herrlinger et al also report an impressive increase in survival within the cohort of patients having a mutated MGMT promoter, the authors indicate that these effects should be interpreted with caution due to small sample size. Furthermore as Stupp points out in the commentary proceding the Herrlinger article in Lancet 2019, these findings are driven by a smaller subset of patients that may have skewed the results of the study.

The above-mentioned issues aside, we understand the reviewer’s comment, and have revised the manuscript to present a more encouraging clinical picture with regard to current treatment of GB and acknowledge the modest, but significant, advancements that have taken place with treatment of GB (see lines 57-62 of the revised manuscript).

Line 62: heparin-binding growth factor…? “HDGF” = hepatoma-derived growth factor

Thank you for pointing that out. It should have been hepatoma-derived growth factor and has been corrected in the revised manuscript (see line 68 of revised manuscript)

Line 67: “GMB” should be GBM

As per the other reviewer's recommendation, GBM has been replaced with GB throughout the manuscript.

Line 287: I assume you refer to Figure 2A (not Figure 1A) here..?

The reviewer is correct we were meaning to refer to Figure 2A. This has been corrected in the revised manuscript (see line 297 of revised manuscript)

Line 287: Abbreviation “WB” needs to be introduced 

WB has been introduced (see line 297 of revised manuscript) 

Fig 2:

Line 296: “fold change” – what does that mean?

The “Fold change” was referring to change in expression compared to the control. For clarity we have replaced “Fold change” with “relative change in SAT1 mRNA expression.” We have also defined the relative change in the Figure legend (see Figure 2, Line 306 in revised manuscript)

Line 296/297: “..after 72: A)..” – after 72 what?

The 72 refers to the time period. The figure caption has been revised to “72 hours”

Line 298: “..knockdown with measured after 72 hours..” – with what?

The figure caption was changed to “SAT1 protein knockdown measured using WB after 72 hours of transfection with LNP-siSAT1

Figure Legend: Explain for what purpose beta-actin was included (2 C), make clear what red fluorescence is (C) and D)), and clarify what part of the Figure Legend for Fig. 2 refers to C) and D)

The figure caption was changed:

“Figure 2. Relative SAT1 mRNA levels in LNP-siSAT1 transfected U251 cells against control which received LNP-siSCR, after 72 hours: A) with and without APOE (1 µg/mL) in media and B) at 40 and 80 nM siRNA concentration (n = 3, p<0.01). C) SAT1 protein knockdown measured using WB after 72 hours of transfection with LNP-siSAT1. β-actin was used as the internal control. Immunofluorescence of SAT1 (green) and DAPI stain for the nucleus (red) in the control (D) and SAT1 KD U251 (E) cells after 72 hours. Values are expressed as the mean ± standard error of the mean (SEM); p < 0.05 was considered statistically significant.”

Fig 3:

Please specify the number of experiments underlying all your cell viability analyses / the actual “n” underlying your statistical analyses.

The values are from three replicates (n = 3). This information has been added to the caption

Explain how cell viability can be higher than 100% (LNP-siSCR in A), B), and C), and LNP-siSAT1 in B)).

Cell viability was determined based on the colormetric MTT assay which measures the formation of a formazan metabolite within the mitochondria of viable cells. The quantitative measure of viability is based on the following:

Absorbance (LNP-siSCR or LNP-siSAT1 treated cells)/Absorbance (cells treated with media alone) × 100.

While the average viability may be higher than 100% in the LNP treatment groups, these values (higher than 100 %) are not statistically greater than observed in control cells that just received media. One of the advantages of the LNP formulations for siRNA delivery is the low cytotoxicity produced compared to viral or non-viral polycationic polymer approaches.

Is there a reason why the order of agents is different in B) compared to A), C) and D)? If not, all plots should be in the same order.

Thank you for pointing this out, the order of agents used in the figure has been revised and is now similar throughout all the cells examined.

I suggest to label all your plots (A-D) with the name of the cell line it represents to improve readability.

The names of the cell lines are now included in the plot, as suggested (see line 334 of revised manuscript).

Fig. 4:

Please specify the number of experiments underlying all your cell viability analyses / the actual “n” underlying your statistical analyses.

The figure caption has been modified and n value was included as suggested:

“Figure 4. Cytotoxic response to different doses of radiation in SAT1 KD U251 cells (A). Values are expressed as the mean ± standard error of the mean (SEM, n = 3). The corresponding calculated CTI values (B) for plot A values; CTI <0.7 indicates a significant synergistic effect.”

Fig 5:

Line 350: should be “fluorescence microscope”

The reviewer’s suggestion has been incorporated into the revised manuscript (see line 370 of revised manuscript)

Line 353: “images E and F are the corresponding nuclei..” – I suppose you mean E and H

The reviewer is correct, the figure caption has been revised to reflect the correct images (see line 373 of revised manuscript)

Since this is already a color figure, I would strongly recommend to show D & G in green and E & H in red. This would immediately make clear that both F and I are overlays.

The images are captured in grayscale and were pseudo-colored only in the merged channel because B/W images displayed better contrast and allows for better visualization. While coloring all panels in red or green would not add or delete to our findings, we would prefer to leave the B/W images. We have revised the figure legend to clarify what the color figures represent in the revised manuscript (see lines 373-375 of revised manuscript)

Please specify the number of experiments underlying all your flux / permeability studies (B) and mRNA level measurements (C) so that the actual “n” underlying your statistical analyses is revealed.

The number of replicates (n =3) has been added to the figure legends.

Line 512: “…that SAT1 is a key factor… in upstream DNA damage repair, and could be targeted using..”

This line was removed as part of the revision.

Discussion:

There should be a thorough, explicit discussion of the study’s limitations, such as the cell line used, the (tumor) cell viability model, the blood brain barrier model etc.

If your findings are supposed to lay the groundwork for actual treatments, i.e., if they are supposed to be relevant, what are the next steps? What challenges need to be overcome until there can be clinical trials in humans?

The reviewer asks a critical question and based on these initial studies, the next steps would be translational studies in a mouse orthotopic xenograph model. Of note, we have already begun identifying suitable patient derived primary GB cells that we could use for this purpose, as confirmation in a patient derived primary cell(s) would provide the most convincing translational pre-clinical data. Our collaborator, Dr. Tanveer Sharief has completed the initial target knock-down validation with our LNPs in two different patient derived primary cells (in vitro). Of note, we see comparable degree of knockdown with the LNPs as we have reported here with the U251. While we are still working through the viability studies both alone and with radiation exposure in the primary patient derived cells we do see a definitive reduction in spheroid formation following SAT1 KD with the LNPs in these primary cells. We are excited about these translational studies but would like to get the initial report of the effects of the LNP-siSAT1 out first.

We have added some discussion on the limitations of the current studies and the important translational studies required to advance this technology (see lines 545 of revised manuscript)

Reviewer 2 Report

In this manuscript, the authors examine the effects of silencing the expression of the mRNA coding for the spermidine/spermine N1-acetyltransferase 1 (SAT1), an enzyme responsible for polyamine catabolism which expression is increased in glioblastoma and correlates with increased resistance to radiotherapy. To do so, they take advantage of a lipid nanoparticle-based siRNA delivery system. In the first part of the manuscript (Fig 1-4), they describe the steps undertaken to improve the quality of the particles (composition of the particles, preparation method, titration to optimize transfection efficiency and silencing) and examine the efficiency of silencing by monitoring the levels of RNA and protein expression in tumor and non-tumor cells. They show that silencing is not occurring in a cell-specific manner, since SAT1 expression is also silenced in non-tumor cells. But the functional effects appear to be tumor cell-specific since only transfected tumor cells are affected in their viability. The authors go on and analyse the effects of silencing SAT1 on DNA damage in the tumor cells (Fig 5) and show that it induces double strand breaks, as indicated by data from the Comet assay and from γ-H2AX phosphorylation. They finally address the capacity of the particles to cross the endothelial membrane in an indirect co-culture system and show that particles can cross these membranes and transfect the tumor cells if a Cadherin peptide is added to facilitate the opening of the endothelial membranes. The authors conclude that the “use of cadherin binding peptides to improve BBB penetration and the delivery of siSAT1-520 LNPs to tumor cells could provide a method for more effective treatment of GBM”.

This is an interesting work, with experiments correctly conducted and, in most cases, correctly interpreted, except in the case of the results related to DNA damage and repair. I have there some major criticisms which I report below.

My comments:

1-Graphical abstract: it is misleading and incomplete. In its current presentation, I would conclude that silencing SAT1 leads to impaired polyamine metabolism and to DNA repair. If the former is expected, it is not demonstrated in the manuscript; regarding the latter, the authors show that silencing of SAT1 triggers DNA damage but there is no demonstration that these damages are repaired. I therefore recommend the authors to redraw a graphic that would better fit their results and conclusions.

2-Abstract: it lacks a clear interpretation of the data and conclusion. The authors should make clear that they report in vitro work. The expression “GBM cell (U251) model” could be misleading. I strongly suggest to replace this expression by “the human glioblastoma cell line U251”. I also kindly recommend the authors to use “glioblastoma” instead of “glioblastoma multiform” which is no longer in use, hence to use the abbreviation GB instead of GBM.

3-Methods:

-The authors should indicate why they use APOE. This would be helpful for readers who are not familiar with the use of nanoparticles.

-There is no mention of the source for the siRNA control and the siSAT1.

4-Results:

Figure 2 legend: (C) and (D) are missing

Fig S1 legend: it starts with a A) that should be deleted

Fig S3 legend: the blot shows two lanes with C; the graph for quantitation shows 2 bars with B. Please explain why these 2 C or 2 B samples and correct appropriately.

-Line 295: here we find the first mention of the LNP(siSCR). The control particle should have been introduced earlier when the authors describe particles preparation for instance.

Figure 5 and related text:

-line 336: the authors write that “the impact on DNA repair was examined using a comet assay”: this is not correct, the Comet assay measures DNA double strand breaks (DSB), not DNA repair.

-line 340: the authors write “…significant reduction in DNA damage repair”. This is not correct, this is an over interpretation of the Comet assay. The data show that silencing of SAT1 induces DNA DSB! It does not mean that the cells have lost their capacity to repair and the authors have not looked experimentally at this capacity.

-lines 344-345: the authors write “γ-H2AX foci can be seen at the vicinity of the DSB in the nucleus (Figure 6).” First, the reference to Figure 6 is wrong, this should be Figure 5. Second, I would recommend to rephrase the sentence. I don’t think that anyone can see a double strand break at that level of magnification!

-line 347: “impaired DNA repair capacity”. The same mistake as commented in line 340 is repeated here.

-line 349: “improved DNA repair in U251 cells”. Why “improved” ? The experiment shows that SAT1 expression is important for DNA repair, but I don’t see data in the manuscript that allows a comparison for an “improved” repair. Please rephrase and/or clarify.

-check the legend to Figure 5; the (B) is not cited.

-It would have been interesting to examine whether SAT1 silencing induces DSB in the non-tumor cells, or not. This might be part of the mechanisms that ensure a specific effect of silencing SAT1 in tumor cells. Maybe the authors could comment in their discussion on that issue.

Figure 6:

-line 364: the authors should briefly introduce again here what ADTC5 is.

-legend: the (A) is not cited.

Discussion:

-lines 425-426: please rephrase the sentence.

-line 442: the authors write “In addition, impaired DNA DSB repair resulted…”. Once again, I see here a misinterpretation or over interpretation of the data obtained in this work. The authors have used methods and experimental schedules that allow them to assess the level of DNA damage, of generation of DSB. They have not addressed whether repair is affected or not. For that purpose, they could have examined this question by performing a kinetic analysis of the foci: disappearance of the foci over time suggests the presence of repair mechanisms. Or they could have analysed the signalling pathways associated to repair. In absence of these data, I would strongly recommend to refrain from talking about “repair”.

-line 504: the authors write “In this proof-of-concept study, we have demonstrated that transient opening of the BBB…” I cannot find the demonstration of a transient opening. The authors should clarify that point.

Finally, I recommend the authors to carefully read-proof their manuscript for typos and spelling mistakes.

Author Response

We thank all the reviewers for their valuable considerations and suggestions, which helped us improve the paper. Please find below our answers to all comments and concerns, point by point. All changes in the manuscript are highlighted in red to allow easy identification.

Graphical abstract: it is misleading and incomplete. In its current presentation, I would conclude that silencing SAT1 leads to impaired polyamine metabolism and to DNA repair. If the former is expected, it is not demonstrated in the manuscript; regarding the latter, the authors show that silencing of SAT1 triggers DNA damage but there is no demonstration that these damages are repaired. I therefore recommend the authors to redraw a graphic that would better fit their results and conclusions.

The concern expressed about the graphical abstract was taken into consideration. The graphical abstract has been redrawn to more closely reflect the experimental findings (see line 51 of revised manuscript).

Abstract: it lacks a clear interpretation of the data and conclusion. The authors should make clear that they report in vitro work. The expression “GBM cell (U251) model” could be misleading. I strongly suggest to replace this expression by “the human glioblastoma cell line U251”. I also kindly recommend the authors to use “glioblastoma” instead of “glioblastoma multiform” which is no longer in use, hence to use the abbreviation GB instead of GBM.

The abstract has been modified as suggested by the reviewer (see lines 28-46 of revised manuscript).

Methods:

The authors should indicate why they use APOE. This would be helpful for readers who are not familiar with the use of nanoparticles.

This has been added to the manuscript (See line 435 of the revised manuscript)

There is no mention of the source for the siRNA control and the siSAT1.

This information has been added to the manuscript (see lines 127-130 of revised manuscript).

Results:

Figure 2 legend: (C) and (D) are missing.

This omission is corrected in the revised manuscript (see lines 306-312 of revised manuscript).

Fig S1 legend: it starts with a A) that should be deleted

This has been changed in the supplementary information.

Fig S3 legend: the blot shows two lanes with C; the graph for quantitation shows 2 bars with B. Please explain why these 2 C or 2 B samples and correct appropriately.

The reviewer’s concerns have been addressed and the revised figure legend more accurately describes the figure.

Line 295: here we find the first mention of the LNP(siSCR). The control particle should have been introduced earlier when the authors describe particles preparation for instance.

The LNP-siSCR has been introduced in the methods section (see lines 147 and 150 of revised manuscript).

Figure 5 and related text: -line 336: the authors write that “the impact on DNA repair was examined using a comet assay”: this is not correct, the Comet assay measures DNA double strand breaks (DSB), not DNA repair.

Yes, we agree that the comet assay measures DNA damage. We have made changes in the results section and added discussion that clarifies the interpretation of the data from the comet assay. (See lines 351-355, 358-366, 466-477 of revised manuscript).

line 340: the authors write “…significant reduction in DNA damage repair”. This is not correct, this is an over interpretation of the Comet assay. The data show that silencing of SAT1 induces DNA DSB! It does not mean that the cells have lost their capacity to repair and the authors have not looked experimentally at this capacity.

In examining the reviewer’s comment, it seems we have not fully explained or provided the rationale supporting our contention that the effects observed following radiation exposure in the SAT1 KD cell group reflects altered DNA repair. Silencing of SAT1 alone did not cause an increase in DNA DSB. If our response was due to an increased propensity towards double strand breaks, we would have anticipated some indication of that even in the cells treated with LNP-siSAT1 alone. We have included the comet assay on control (media only) and SAT1 KD cells that were not exposed to radiation in the supplementary information to improve clarity. SAT1 KD alone did not produce observable tail moment in the comet assay.

Additional information supporting a reduction in DNA DSB repair can be found with the temporal nature of the DSB observed (included in the supplemental data). When SAT1 is silenced, there is an abundance of DSBs detected at 6 hours following radiation exposure, as visualized by the tail moment and surrogate marker g-H2AX, indicating either ablation or at least a temporal delay in DSB repair. Compared to the comet assay on control and SAT1 KD cells 24-hours after exposure to radiation, where the tail moments were not as pronounced as seen at 6 hours, this could be a combination of loss of cell population with DNA damage or completion of DNA damage repair in surviving cell population. These findings together with the previous published studies demonstrating SAT1 and its correlation with DNA repair pathway suggests the effects we see are due to altered DNA DSB repair.

We have addressed this in the revised manuscript (see lines 351-355, 358-366, 466-477 of revised manuscript)

lines 344-345: the authors write “γ-H2AX foci can be seen at the vicinity of the DSB in the nucleus (Figure 6).” First, the reference to Figure 6 is wrong, this should be Figure 5. Second, I would recommend to rephrase the sentence. I don’t think that anyone can see a double strand break at that level of magnification!

We have correctly identified Figure 5 as the figure containing the γ-H2AX foci in the revised manuscript. We appreciate what the reviewer is indicating with the statement regarding visualization of DSB at this level of magnification. We note in the revised manuscript that γ-H2AX is widely used as a surrogate marker for DNA DSB visualization. γ-H2AX foci are known to form close to the vicinity of the DSB as a part of DNA damage repair pathway. Visualizing γ-H2AX foci is an indicator for presence of DSB. We have included γ-H2AX immunofluorescence image on control and SAT1 KD that was exposed to higher 10 Gy radiation in the supplementary information (Figure S4 E and F). The presence of higher number of foci in 10 Gy radiation compared to 1 Gy radiation (Figure 5 F and I) is an indication that there higher number of DSB in 10 Gy group.

line 347: “impaired DNA repair capacity”. The same mistake as commented in line 340 is repeated here.

This question has been addressed above comment on “line 340”.

line 349: “improved DNA repair in U251 cells”. Why “improved” ? The experiment shows that SAT1 expression is important for DNA repair, but I don’t see data in the manuscript that allows a comparison for an “improved” repair. Please rephrase and/or clarify.

The discussion has been rephrased in the manuscript to reflect the reviewer’s concerns (see lines 351-355, 358-366, 466-477 of revised manuscript)

check the legend to Figure 5; the (B) is not cited.

This has been fixed.

It would have been interesting to examine whether SAT1 silencing induces DSB in the non-tumor cells, or not. This might be part of the mechanisms that ensure a specific effect of silencing SAT1 in tumor cells. Maybe the authors could comment in their discussion on that issue.

We understand the reviewer’s query regarding DSB in the non-GB cells. As we observed no overt toxicity with SAT1 silencing alone and no improved cytotoxicity response to radiation or chemotherapy in the non- GB cells, any effects of DSB were not sufficient to produce toxicity and for this reason we did not examine DSB in those cells. We note the DSB even in the U251 without radiation exposure was not influenced by SAT1 silencing.

Figure 6:

line 364: the authors should briefly introduce again here what ADTC5 is.

ADTC5 has been introduced as suggested:

The cadherin binding peptide ADTC5 treated hCMEC/D3 monolayers displayed a 2.8-fold higher flux of IRdye-PEG compared to the control cells.”

legend: the (A) is not cited.

The figure legend has been corrected in the revised manuscript (see line 393)

Discussion:

lines 425-426: please rephrase the sentence.

As per reviewer’s suggestion we have revised the sentence addressing the polyamines and their role in tumor proliferation and growth

“Polyamines such as spermidine and spermine play essential roles in cell functions, including maintaining chromatin structure, facilitating growth, proliferation of the cell, regulating ion channels, and scavenging free radicals”

line 442: the authors write “In addition, impaired DNA DSB repair resulted…”. Once again, I see here a misinterpretation or over interpretation of the data obtained in this work. The authors have used methods and experimental schedules that allow them to assess the level of DNA damage, of generation of DSB. They have not addressed whether repair is affected or not. For that purpose, they could have examined this question by performing a kinetic analysis of the foci: disappearance of the foci over time suggests the presence of repair mechanisms. Or they could have analysed the signalling pathways associated to repair. In absence of these data, I would strongly recommend to refrain from talking about “repair”.

As per the reviewer’s suggestion we have included the kinetic analysis of the DSB data following radiation exposure and our interpretation of this data to support our claim regarding repair mechanisms being involved in the response we are observing with LNP-siSAT1. The comet assay and γ-H2AX immunofluorescence were used as surrogate markers of DSB in control and SAT1 KD cells post irradiation. After 24 hours, once the DNA DSB repair is complete the comet assay tail was not observed and γ-H2AX are no longer recruited and cannot be visualized.

Hence based on longer tail moment in comet assay and higher γ-H2AX foci in SAT1 KD cells 6 hours after irradiation, we conclude that there is more DSB in SAT1 KD group. Hence, it is an indication of impaired (reduced or delayed) DSB repair in SAT1 KD cells compared to control cells.

line 504: the authors write “In this proof-of-concept study, we have demonstrated that transient opening of the BBB…” I cannot find the demonstration of a transient opening. The authors should clarify that point.

Here a hydrophilic IR dye was used as a paracellular (intercellular) permeability marker. The higher flux of the marker in ADTC5 treatment group compared to the control would indicate transient opening of the BBB model. This has been explained in the results section:

“The ADTC5 treated hCMEC/D3 monolayers displayed a 2.8-fold higher flux of IRdye-PEG compared to the control cells. The higher flux of the macromolecule permeability marker indicated that the ADTC5 disruption was successful, thereby enabling the paracellular diffusion of the hydrophilic dye.”

Finally, I recommend the authors to carefully read-proof their manuscript for typos and spelling mistakes.

Thank you for the recommendation, we have proof-read the revised manuscript for grammar and spelling.

Reviewer 3 Report

In this study Yathindrannth et al studied siRNA-mediated knockdown of SAT1 gene in GBM cell line U251. The authors formulated an ionizable lipid nanoparticle based approach to deliver the siRNA, and demonstrated that knockdown of SAT1 sensitized U251 cells to radiation, but did not have sensitization effects on several non tumor cell types. The authors further used an in vitro BBB permeability assay models, involving hCMEC/D3 as BBB barrier, to show that in the presence of the ADTC5 peptide, the siRNA could be delivered to U251 cells, resulting in measurable SAT1 gene knockdown. Formulating nanoparticle-based siRNA knockdown and devising strategies for their delivery to overcome the BBB is an important area of research and can be potentially interesting. Several major limitations are noted and should be addressed:

(1) The findings from the SAT1 targeting should be confirmed in another cell line, instead of relying on only U251 cell line.

(2) In Fig. 5 experiments, cells (control or SAT1-kd) without radiation need to be included as non radiation control in order to more accurately assess their response to radiation. In addition, the timing of recovery should be considered: for example, had the control cells mostly recovered from the radiation at 6 hours post radiation such that there were fewer foci?(Fig. 5J)  

(3) The presentation of results had numerous errors and was confusing, including in absence of figures or legend labels in multiple places, e.g., in Fig. 2, 5, 6. In Fig S3, it was indicated a whole gel image for Fig. 2C, however Fig. 2C was an IF image which was never been described in legends and in the manuscript; in addition in the same Fig. S3, the sample's labels (ABCC and ABBC) were confusing.

(4) In the case of using ADTC5 - it is important to make sure that 1 mM of ADTC5 did not have toxicity on the hCME/D3 cells. In addition, the SAT knockdown in the BBB assays was not nearly as potent as previous experiments - was ApoE used in this assays? Was this seemingly low knockdown efficacy also due to the low BBB permeability in the assay? This question may be addressed by examining the SAT1 knockdown in the hCME/D3 cells that were used as the BBB in the assays. In addition, was this level of knockdown similarly affecting tumor cell's viability and their sensitivity to radiation?

Author Response

We thank all the reviewers for their valuable considerations and suggestions, which helped us improve the paper. Please find below our answers to all comments and concerns, point by point. All changes in the manuscript are highlighted in red to allow easy identification.

The findings from the SAT1 targeting should be confirmed in another cell line, instead of relying on only U251 cell line.

We understand reviewer’s concern about confirmation of the effects of SAT1 silencing in additional GB cell lines. While we used LNP formulations to deliver the siRNA to silence SAT1– previous studies have shown similar sensitization responses following SAT1 KD in various GB cell lines. These studies (Thakur et al. Oncogene (2019) 38:6794–6800) certainly fit with our data. As mentioned previously, we are actively pursuing the LNP-based silencing of SAT1 in patient derived primary GB cells. We are able to reduce levels of SAT1 in two different SAT1 patient derived primary cell culture models. We are currently evaluating the effects on cell viability and response to radio and chemotherapy. These studies are still in process, but we do not a definite change in the morphology of the cells and a reduced ability to form spheroids following SAT1 silencing. These suggest to us that SAT1 KD may reduce stem-cell like properties which would also be of interest. Our intent is to move forward with these patients derived primary cells both in vitro and murine xenographs to show the potential therapeutic benefits of this approach. We are encouraged by these ongoing studies but would like to get our initial observations with this LNP formulation out in the literature to allow us the time to fully develop the translational studies with these nanomedicine-based approaches to SAT1 KD in GB.

In Fig. 5 experiments, cells (control or SAT1-kd) without radiation need to be included as non radiation control in order to more accurately assess their response to radiation. In addition, the timing of recovery should be considered: for example, had the control cells mostly recovered from the radiation at 6 hours post radiation such that there were fewer foci?(Fig. 5J)

The comet assay control cells that were not exposed to radiation was added to the supplementary information (Figure S4 A and B). Post irradiation considerable DNA damage was observed at 3 and 6 hours. However, by 24 hours a complete recovery of DNA damage was observed (No tail in comet and no observed foci).

The presentation of results had numerous errors and was confusing, including in absence of figures or legend labels in multiple places, e.g., in Fig. 2, 5, 6. In Fig S3, it was indicated a whole gel image for Fig. 2C, however Fig. 2C was an IF image which was never been described in legends and in the manuscript; in addition in the same Fig. S3, the sample's labels (ABCC and ABBC) were confusing.

Yes, we agree. There was an error with the files that got uploaded in the submission portal. We had requested the editorial team to fix it with the latest files before review process. However, the pdf file was updated, and the word file was not it seems. Thank you for your patience and understanding in this regard. We have corrected all these errors now.

In the case of using ADTC5 - it is important to make sure that 1 mM of ADTC5 did not have toxicity on the hCME/D3 cells. In addition, the SAT knockdown in the BBB assays was not nearly as potent as previous experiments - was ApoE used in this assays? Was this seemingly low knockdown efficacy also due to the low BBB permeability in the assay? This question may be addressed by examining the SAT1 knockdown in the hCME/D3 cells that were used as the BBB in the assays. In addition, was this level of knockdown similarly affecting tumor cell's viability and their sensitivity to radiation?

ADTC5 is a well-established pharmacological BBB opener that our group has been working on for nanoparticle and drug delivery. The ADTC5 at 1 mM concentration was nontoxic to hCMEC/D3 cells, and APOE was added along with the LNPs to aid transfection of the GB cells. As the purpose of this experiment was to provide proof-of-concept for the delivery approach across a BBB model we were focused on demonstration of knockdown of target. The confirmation of sensitivity of the SAT1 KD cells to radiation and cytotoxicity studies were not performed.

Reviewer 4 Report

In this paper the authors prepared a lipid nanoparticle-based siRNA delivery system (LNP-siSAT1) to selectively knock-27 down (KD) SAT1 enzyme in a GBM cell (U251) model. The work is well done, and the conclusions reached are consistent with the experimental results

Author Response

We thank all the reviewers for their valuable considerations and suggestions, which helped us improve the paper. Please find below our answers to all comments and concerns, point by point. All changes in the manuscript are highlighted in red to allow easy identification.

In this paper the authors prepared a lipid nanoparticle-based siRNA delivery system (LNP-siSAT1) to selectively knock-27 down (KD) SAT1 enzyme in a GBM cell (U251) model. The work is well done, and the conclusions reached are consistent with the experimental results

Thank you for the kind feedback.

Round 2

Reviewer 1 Report

The authors have addressed my concerns.

Author Response

Dear Editor,

We thank you and the reviewers for the consideration and valuable suggestions, which helped us to improve the paper.

We agree with you about the need for an additional GB cell line. In this revised manuscript, in addition to U251 cells, we have repeated the studies in two other GB cell lines (LN229 and 42MGBA). We have looked at the SAT1 knockdown and established the sensitization towards GB cells. We have included additional data from these cell lines in Figures 2F and 2G (SAT1 knockdown), 3E and F (Radiation sensitization), 5K-N (comet assay), and the whole western blots in the supplementary information (Figures S4 and S5). The related additions/modifications in the text are highlighted for convenience.

Reviewer 2 Report

I do thank the authors for their clear explanations and replies to my questions and their efforts to improve the manuscript.

I do not have further concern and am glad to support the publication of the revised manuscript in Cancers.

Author Response

(The authors gave the same response as above.)
